# Do the Age and Gender of Chairperson Affect Firms’ Environmental Protection Investment?

**DOI:** 10.3390/ijerph192114495

**Published:** 2022-11-04

**Authors:** Libin Han, Chong Peng

**Affiliations:** 1Economic and Social Development Research Institute, Dongbei University of Finance and Economics, Dalian 116025, China; 2School of Economics, Nanjing Audit University, Nanjing 211815, China

**Keywords:** chairperson characteristics, environmental investment, political connections, corporate social responsibility motivation, action, and reputation

## Abstract

Environmental investment of companies can not only bring profits to firms but also contribute to environmental protection. However, little is known about how chairperson characteristics affect companies’ decisions on environmental investment. This paper fills the gap in the literature by studying the impact of chairperson characteristics on firms’ environmental investment. For empirical evidence, we conduct a regression on environmental protection disclosure data of Chinese listed companies sourced from the CSMAR database. We find that China’s overall environmental protection disclosure level is relatively low. The age of the chairperson has an inverted U-shaped relationship with environmental investment, and a female chairperson has a significant positive effect on environmental investment. Contrarily, the education level and political connections of the chairperson have no significant impact on firms’ environmental investment. Understanding the effect of these factors will help companies plan their environmental protection activities more efficiently.

## 1. Introduction

In the past decades, China has witnessed rapid economic growth, but the development has been accompanied by severe environmental pollution. The mean PM2.5 was 33 μg/m^3^ in China in 2020, compared to 10 μg/m^3^ in the USA. Environmental pollution in China, which includes air, land, and water pollution, has caused serious health problems. Especially regarding air pollution, one study found that the coal burning policy in China has led to a decrease in life expectancy [1]. The researchers found that the total suspended particulates air pollution causes the 500 million residents of Northern China to lose more than 2.5 billion years of life expectancy. The pollution problem is even more serious in industrialized cities. People are willing to pay for air quality improvement [2].

The government has been well aware of the seriousness of environmental pollution and has acted accordingly. In recent years, the central and local governments have continuously increased their supervision of and investment in environmental protection. In absolute values, investment increased during 2007–2017, from less than 100 billion RMB (Chinese yuan) in 2007 to 600 billion RMB in 2017. The proportion of China’s environmental expenditure to its government fiscal expenditure has also remained stable at 2–3 percent (as shown in Figure 1). In addition, since 2016, the central government has conducted environmental monitoring movement every year to reduce the level of environmental pollution in China. However, despite environmental pollution being a highly negative behavior, companies and local governments lack the motivation to take concrete steps for environmental governance and related investment. Cai et al. [3] pointed out that environmental emission reduction in 2001 triggered an increase in sewage treatment in the counties located at lower reaches of a major river, and its pollution charges were relaxed. The act of “beggar-my-neighbor” has made environmental governance more difficult. For enterprises, stringent environmental regulations increase their production costs and affect their exports [4].

However, green development of the economy implies that economic growth and environmental protection are closely related rather than opposite concepts [5]. Thus, not all companies avoid their environmental responsibility. There are some that actively participate and invest in environmental protection. Entrepreneurs also happily display to the public that their company is actively involved in environmental protection. The obvious motivation for companies to participate in environmental protection is to pursue the benefits of their business. Environmental protection is corporate social responsibility (CSR). Some countries, like the USA, the UK, and India, have enacted CSR laws encouraging positive corporate environmental awareness. China’s goal of achieving carbon peak by 2030 and carbon neutrality by 2060 is inseparable from the efforts of enterprises.

The literature focuses on the impact of corporate governance structure, corporate characteristics, and other factors on corporate environmental protection investment. However, there is little research on the impact of company chairpersons on environmental protection decisions. This study empirically examines the impact of the chairperson’s personal characteristics on the company’s environmental behavior using data from Chinese listed companies. We consider different types of environmental protection behaviors and their impact on the environmental protection investment of enterprises.

We found that the age of the chairperson has inverted U-shaped relationship with environmental investment. Female chairperson increases the investment in the environment. The policy connection has no effect on a firm’s environmental investment.

This paper is organized as follows. Section 1 presents the literature review, Section 2 introduces firms’ environmental investment in China, Section 3 presents methodology and material, Section 4 shows the empirical results, Section 5 is the discussion, and Section 6 concludes.

## 2. Literature Review

Our paper is related to the literature on corporate social responsibility (CSR). Traditional economic theory holds that the core aim of enterprise production is to maximize interest rates or shareholder value. CSR provides more dimensions of corporate value [6]. Many firms are concerned about their CSR activities because of the incentives they bring. The motivation of firms for environmental protection investment can be divided into internal and external factors. Regarding internal factors, enterprises have a certain social responsibility, which is often based on their own interests. Gao et al. [7] found that charitable donations of private enterprises are used more to cover up or divert public attention from their low levels of employee compensation and benefits and their environmental impact, rather than to determine the organic composition of CSR. Thus, they are more likely to use CSR as a profit-making tool. Regarding external factors, outside stakeholders motivate firms to undertake CSR activities [8]. Rodrigue et al. [9] found that pressure from various stakeholders has a significant impact on a firm’s environmental strategy. The research in China found that foreign companies have a positive impact on Chinese companies’ CSR through the supply chain pressure channel, and this effect is significant only when Chinese companies are suppliers of foreign companies [10]. Corporate investment in social responsibility also has an impact on corporate behavior. CSR shapes firm decisions to a certain extent. To improve future financial performance, firms will undertake CSR expenditures in the current period [11]. More socially responsible firms generally receive more favorable news reportage. If socially responsible firms commit to a high standard of transparency and engage in less bad news hoarding, they will have lower crash risk [12]. Large firms and firms with high financial constraints are more motivated to increase their immediate CSR activities following natural disasters [13].

Environmental protection is perhaps the key component of CSR. Policies, past environmental investments, importance of environmental technologies for customers, and firm performance affect firms’ environmental investment [14]. There is higher environmental performance in firms with higher board independence and a lower proportion of directors appointed after the CEO on the board of directors. Firms with greater board gender diversity are less often sued for environmental infringements [15]. Huang and Chang [10] concluded that perceived advancement in technology from FDI reduces environmental pollution in China. Cai et al. [16] found that firms prefer not to damage public relations by hiring as a director someone working at another company with a low social reputation, and social reputation plays an important role in promoting CSR.

Environmental investment benefits firms too. Environmental management may play a pivotal role in firms’ financial performance [17]. Positive relationships have been found between energy efficiency, environmental performance, and firm productivity [18]. In the short term, environmental investment does not affect firm performance significantly, however, in the long term, environmental investment increases firm performance significantly [19]. In order to carry out its environmental responsibility, a firm makes environmental expenditures to reduce the perceived severity of its pollution emissions [20,21]. Banerjee and Gupta [22] used a dataset covering 42 countries spanning 2002–2013 found that firm-level environmentally sustainable practice has a positive effect on R & D. The environmental protection tax “forces” companies to increase their R & D investment [23]. He et al. [24] found that the Environmental Protection Tax Law that was implemented by China in 2018 promotes the TFP of heavily polluting firms.

Our paper is also related to the literature on environmental pollution in China. Chinese listed companies generally make insufficient investments in environmental protection. Environmental protection investment is a kind of “passive” behavior for enterprises and often fails to meet government environmental regulation requirements. Large environmental protection funds have been invested in non-heavily polluting industries [25]. Li et al. [26] found that a company’s environmental protection activism is driven by the opportunist motives of its policymakers, whose goal is to mask their inabilities. Public appeal tends to have a positive effect on green investment in China [27].

Additionally, our paper is related to the literature on corporate governance. Malm et al. [28] found that firms with older CEOs face fewer lawsuits. Firm CEOs have significant effects on firm performance. For example, overconfident managers overestimate the returns to their investment projects, view external funds as unduly costly, and distort corporate investment [29,30].Cassell et al. [31] provided empirical evidence suggesting that CEOs with large inside debt holdings prefer less risky investments and financial policies. Amore et al. [32] found that CEO education significantly improves a firm’s energy efficiency. Francoeur et al. [33] based on a sample of 5222 U.S. firm-year observations, found that such CEOs positively influence environmental performance and that this effect is more prevalent in profitable firms. This result suggests that powerful CEOs are influential in creating sufficient resources to enhance their firms’ environmental performance. Yang et al. [34] found that the CEO’s reputation from donations reduces the probability of their forced turnover. Firms with highly educated CEOs are likely to engage in environmental protection spending activities [35]. CEOs’ given names containing moral meanings enhance corporate investment decisions on environmental protection [36].

Gender has also been found to make a difference. Welsh et al. [37] studied firm performance for women entrepreneurs. They found a positive relationship between women entrepreneurs’ human capital and firm performance. Audretsch et al. [38] found that women-led firms internationalize when corruption is institutionalized, and the outcomes of corrupt behavior are more predictable. Firms led by female CEOs outperform those led by male CEOs in China [39]. Women on corporate boards take actions that reduce risk-taking in firms [40].

Despite the impact that manager behavior has on enterprises, the literature has rarely discussed the role of the manager in environmental protection decisions. Our study attempts to fill this gap and provide related evidence. Based on the literature, we have the main hypothesis: The age and gender of the chairperson may influence CSR.

## 3. Firms’ Environmental Investment in China

The corporate social responsibility (CSR) report data used in this study were sourced from the CSMAR database (www.gtarsc.com, accessed on 8 October 2022), which includes rich data on Chinese listed companies. CSRs include two parts: Basic information and the CSR schedule. Basic information includes statistical time, protection of shareholders’ rights and interests, creditors’ rights and interests, employees’ rights and interests, supplier rights, and consumer rights. In addition, basic information also includes decisions on whether to disclose details of environmental protection and sustainable development, public relations and social welfare, social responsibility system construction, and improvement measures. The other part of SCR includes details of major social responsibility expenditures, such as environmental protection investments and donations for energy saving. The quality of CSRs reports in China has been improved continuously. The Ministry of Ecology and Environment is actively implementing relevant policies and strengthening enterprise environmental information disclosure.

As can be seen from Table 1, the publication of CSR reports began in 2006. The total number of listed companies that year was 1458, and only 21 of them, accounting for less than 2 percent of the total, published CSR reports. Since then, the number of companies publishing CSR reports has steadily increased. By 2016, it had increased to 807. The proportion of companies that have published CSR reports after 2010 has also stabilized at around 25 percent. In 2020, this share was 26.9% of the listed firms.

More than 96 percent of the companies publishing CSR reports disclose environmental protection and sustainable development information (Table 1), with only a handful refraining from environmental protection information disclosure. This shows that companies attach significant importance to environmental protection information disclosure.

Table 2 shows that though the industry distribution of firms publishing CSR reports is wide, 70 percent of these firms are concentrated in 20 industries. Among them, the industries that disclose the most environmental protection information are electronics and computer manufacturing, real estate, and chemical and pharmaceutical industries. Some of the observed firms are in polluting industries, while others, such as those in retail and capital services, may not be seriously polluting.

Interestingly, the content of environmental protection and sustainable development is diverse. It includes both investments in production activities and environmental protection measures unrelated to production. For example, practices such as environmental protection training, energy conservation, green publicity, and others significantly differ from the environmental protection regulations related to firm production. Moreover, businesses do not publish CSR reports every year. As shown in Table 3, we found that the environmental protection and sustainability information disclosed by most companies in our sample, which directly constitutes environmental protection investment, makes up for less than 30 percent of the overall disclosures per year. More than 70% of environmental protection disclosures are simple environmental activities such as video conferencing, electronic billing, and water conservation. There are also significant differences between enterprises in the annual investment in environmental management funds. In practice, some companies invest in very small environmental protection funds, while others invest in large ones.

As shown in Figure 2, It can be seen from the regional spatial distribution of enterprises investing in environmental protection that these company locations are located mainly in relatively large cities such as Beijing, Shanghai, Shenzhen, Tianjin, and Guangzhou. It is possible that the willingness for environmental protection may be caused by the listed companies’ presence in these regions. On the other hand, it may be that these areas have a strong sense of environmental protection.

Next, we empirically examine the impact of the chairperson’s personal characteristics on corporate environmental investment decisions.

## 4. Methodology and Material

The empirical sample data of this paper include mainly the SCR reports and the chairperson’s personal information. For data regarding the corporate manager, we consider the company’s chairperson, not the CEO, because, in China, the chairperson is the ultimate decision-maker for a business unit. The data are sourced from the CSMAR database. The sample size before 2010 was small, so we dropped these companies from the sample and only kept the ones for 2010–2017. The final data sample is 4529.

In the identification strategy, we divide the research problem into two parts. First, we examine the factors that determine the company’s type of environmental investment, which we divide into two types: Direct and indirect investment. Following the description in Section 3, we found that the number of enterprises actually needing funds to invest in the production process is not high. Enterprises have a variety of environmental protection actions as options. Therefore, do chairperson’s personal characteristics affect the type of environmental investment? In order to determine this effect, we analyzed the company’s direct investment, which is related to production, and its indirect investment, which is unrelated to production. For example, we classified as an indirect environmental investment the amount of electricity saved, as disclosed under the company’s environmental information. We used the following probability selection model to test this effect.
(1)$$Type_{i}=\alpha +\sum \beta_{i}X_{i}+{\sum \gamma_{i}Z}_{i}+t+\varepsilon_{it}$$

$Type_{i}$ is a 0–1 dummy variable, 1 indicates the company’s indirect environmental protection investment, and 0 indicates the direct environmental protection investment. $X_{i}$ is a personal characteristic of the chairperson, which covers age, gender, educational level, and political connection. We focus on the age and gender of the chairperson. $Z_{i}$ includes corporate-level features, including returns on assets (ROA), firm age, and firm size. As most of the company’s environmental protection investment behavior is non-continuous, such investment is relatively volatile. Moreover, some environmental protection inputs have long-term effects. For example, if an enterprise has invested in a certain year, it may reduce the investment in the following years. There is less scope for a company to change its chairperson in the short term. Therefore, the probability selection model used in this study does not control for firm fixed effects.

On the basis of the above, we further examine whether the personal characteristics of the chairman affect direct investment?
(2)$$Inv_{i}=\alpha +\sum \beta_{i}X_{i}+{\sum \gamma_{i}Z}_{i}+t+\varepsilon_{it}$$

Similar to model (1), we used model (2) to test the effect of chairperson characteristics on direct environment investment. $Inv_{i}$ indicates the direct environmental protection investment.

## 5. Results and Discussion

### 5.1. Statistical Tests

The descriptive statistics of the data in the regression are given in Table 4.

### 5.2. Results Analysis

Table 5 reports the results of the regression. In column (1), we control for the chairperson’s personal characteristics, column (2) controls for characteristics of enterprises, and in column (3), we control for political connections of entrepreneurs. From the regression results in columns (1)–(3), as the chairperson’s age increases, there will be an inverted U-shaped effect on the firm’s selection of environmental investment type: Older chairpersons tend more toward direct environmental investment, but after a certain age, they tend toward indirect investment. This turning point is at around 54 years of age (column (3)). In addition, compared with chairmen, chairwomen are more inclined toward indirect environmental investment. Even if the number of chairwomen is very low (3 percent), the effect is still significant. We also considered the impact of political connections. Some of the chairpersons are also in various administrative positions, such as in the CPPCC. Such political identity may affect the company’s environmental investment decisions. From column (3), we see that this effect is not significant, which means that the company does not determine the amount of indirect environmental protection investment based on the chairperson’s political identity. Another important factor affecting the chairperson’s environmental awareness is their level of education. In listed companies, chairpersons are generally highly educated, with most having undergraduate or higher degrees. We classified chairpersons with college or university degrees as those with low academic qualifications, and chairpersons with university degrees, or higher, as those with high academic qualifications. The education information of entrepreneurs is not comprehensive, and, thus, we lost some variables after individually controlling for the characteristics of chairperson education levels in columns (4)–(5). The regressions in columns (4)–(6) of Table 5 indicate that the education level does not significantly affect what type of environmental investment companies choose.

Some significant effects can be seen at the enterprise level. Larger and older companies are less inclined to disclose information on indirect environmental protection investment. This means that the larger the company, the more it tends toward direct environmental protection investment. There is no significant correlation between the profitability of the company and its indirect environmental protection investment.

Table 5 shows that the chairperson’s age and gender are key factors influencing the choice of type of environmental protection investment.

Table 5 shows that the chairperson’s age and gender can influence a company’s environmental decision-making. Do the personal characteristics of the chairperson affect direct investment? Table 6 shows the analysis results of environmentally friendly companies that actually invested in production. The variables in columns (1)–(3) are the logarithm of the company’s environmental investment funds, and the proportions are given in columns (4)–(6). As the regression results show, the direct environmental protection investment of enterprises will increase first and then decrease as the chairperson’s age increases. This means that older chairpersons will reduce their spending on direct environmental protection. However, columns (4)–(6) indicate that as the chairperson’s age increases, environmental protection investment in total assets increases. The effect of gender is not significant in columns (4)–(6). This means that although women will significantly reduce direct environmental protection investment, the difference in environmental protection investment in total assets is small. The impact of political connections is not significant here either. Firm size and ROA have a significant effect on the amount of direct environmental investment but no effect on the ratio of total assets.

Table 6 shows that the chairperson’s age is an important factor affecting a firm’s environmental investment, and it has an inverted U-shaped effect.

In Table 2, the industry distribution of firms’ environmental protection investment shows that some industries attract more investment. If the industry distribution of firms correlates to chairperson characteristics, our empirical results in Table 5 and Table 6 may have the omitted variable bias. To resolve this problem, we control the industry-fixed effects in Table 7. Because the number of firms is small in some industries. We classify industries into secondary and non-secondary. Some industrial enterprises create pollution during production, so they need to invest in environmental protection. Some others do not cause pollution, so their environmental protection behavior may be reflected more in non-productive inputs. Table 7 reports the regression results. We observe that the coefficient of chairperson age and gender are still significant after controlling the industry-fixed effects. When industry dummy variables are controlled, non-industrial enterprises will choose a more flexible approach toward environmental investment. However, for those firms that actually invest in production, the differences across industries are not significant.

### 5.3. Discussion

This study examines the impact of the chairperson’s personal characteristics on CSR. We distinguished the heterogeneity effect between the company’s direct and indirect environmental protection investments. We found that chairpersons are more inclined to increase indirect environmental investment. The age and gender of the chairperson are important factors influencing such decisions. We also find that political connections do not significantly increase a company’s environmental responsibility information disclosure. In addition, the company size is an important factor determining environmental investment: Large companies are generally more environmentally responsible and aware.

Our paper has some limitations. This paper focuses on objective characteristics like age and gender, which may not fully reflect the personal characteristics of enterprise managers. We also did not explore the underlying reasons behind gender differences in this paper. Due to data limitations, we do not discuss the trend after 2017. From 2017, the central government and local government in China strengthened guidance on firm environmental disclosures, these may affect the behavior of the firm.

In further research, more subjective and objective indicators about the chairperson may help understand the firm environment investment. The influencing factors of gender differences are also worth studying.

## 6. Conclusions

Overall, this paper found that most chairpersons lack the motivation to invest in environmental protection equipment. In the future, governments should formulate policies to motivate companies to participate in environmental protection. Encouraging a gender-equitable entrepreneurial environment may contribute to environmental protection.

## Figures and Tables

**Figure 1 ijerph-19-14495-f001:**
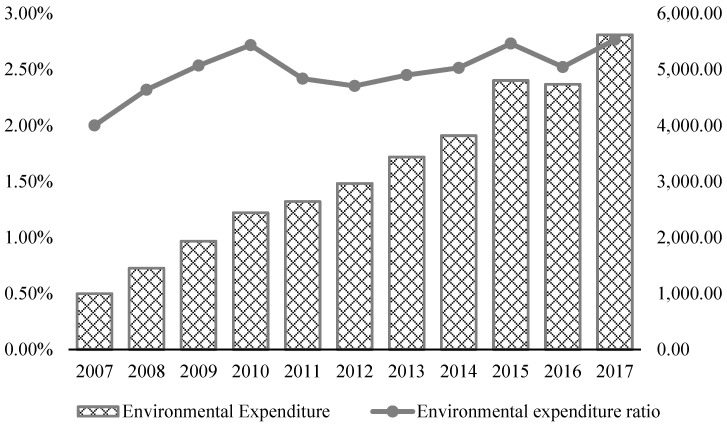
Environmental protection investment in China during 2007–2017.

**Figure 2 ijerph-19-14495-f002:**
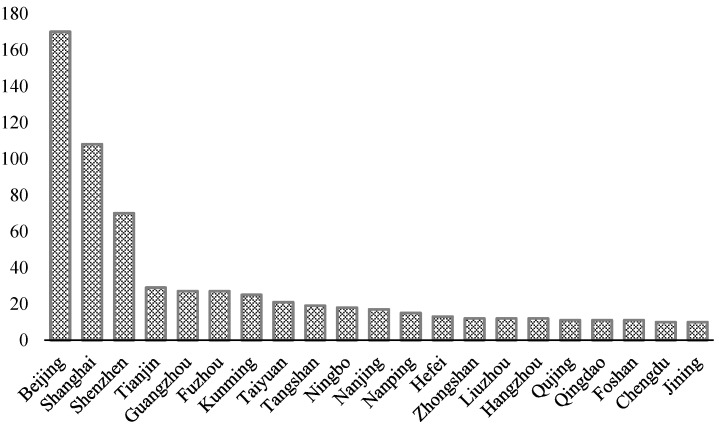
Distribution by cities.

**Table 1 ijerph-19-14495-t001:** CSR disclosure and environmental investment.

Year	List Numbers	Social Report	Social Report Ratio	Environment Report	Environment Report Ratio
2006	1458	21	1.44%	21	100.00%
2007	1572	42	2.67%	41	97.62%
2008	1626	178	10.95%	178	100.00%
2009	1774	186	10.48%	184	98.92%
2010	2127	501	23.55%	491	98.00%
2011	2366	590	24.94%	584	98.98%
2012	2494	653	26.18%	642	98.32%
2013	2543	683	26.86%	662	96.93%
2014	2651	712	26.86%	685	96.21%
2015	2842	759	26.71%	730	96.18%
2016	3136	807	25.73%	780	96.65%

**Table 2 ijerph-19-14495-t002:** Industry distribution of firms (the top 20 industries).

Industry	Numbers
Manufacture of communication equipment, computers, and other electronic equipment	336
Real estate	289
Manufacture of chemical raw materials and chemical products	267
Manufacture of medicines	248
Manufacture of electrical machinery and equipment	206
Production and distribution of electric power and heat power	205
Special purpose equipment	199
Smelting and pressing of nonferrous metals	194
Nonmetal mineral products	146
Capital markets services	141
Software and information technology services	139
Manufacture of general-purpose machinery	134
Manufacture of automobiles	130
Retail trade	126
Monetary and financial services	125
Smelting and pressing of ferrous metals	121
Building projects	117
Manufacture of alcohol, beverages, and refined tea	105
Wholesale trade	103
Mining and washing of coal	101

**Table 3 ijerph-19-14495-t003:** Amount of environmental investment.

Year	Firms	Ratio	Mean (Yuan)	Min	Max	SD
2006	3	14.29%	5433.333	700	13,200	6779.626
2007	6	14.63%	1847.533	121.6	6673	2522.291
2008	28	15.73%	30,404.93	6.6	403,300	80,106.41
2009	35	19.02%	19,767.74	7.5	159,400	42,921.73
2010	116	23.63%	773,035.6	1.6	6.47 × 10^7^	6,346,402
2011	154	26.37%	263,390.6	0.18	3.27 × 10^7^	2,647,539
2012	166	25.86%	356,695	2.7	3.33 × 10^7^	2,891,349
2013	189	28.55%	215,354.3	0.972	1.16 × 10^7^	1,267,599
2014	176	25.69%	206,798.2	0.8	2.21 × 10^7^	1,703,450
2015	179	24.52%	810,860	1	7.18 × 10^7^	6,230,793
2016	176	22.56%	5,912,229	1.48	8.81 × 10^8^	6.47 × 10^7^

**Table 4 ijerph-19-14495-t004:** Descriptive statistics for the regression sample.

Variable	Obs	Mean	Std. Dev.	Min	Max
**Chairman Characteristics**					
Age	4529	53.10	6.47	23.00	79.00
Female	4529	0.03	0.18	0.00	1.00
Education	3493	3.67	0.86	1.00	6.00
Political connection	4529	0.10	0.30	0.00	1.00
**Firm characteristic**					
Size	4529	2.96 × 10^10^	1.27 × 10^11^	36,300,000	2.79 × 10^12^
ROA	4529	0.05	0.06	−0.64	0.67
inv_type	4529	0.75	0.44	0.00	1.00
Inv	4529	320,602.7	1.29 × 10^7^	0	8.51 × 10^9^
Firm age	4529	15.69	5.37	0.00	36.00

**Table 5 ijerph-19-14495-t005:** Chairperson characteristics and environmental investment type.

	(1)	(2)	(3)	(4)	(5)	(6)
Variables						
Age	−0.143 ***	−0.0801 **	−0.0800 **	−0.197 ***	−0.117 **	−0.117 **
	(0.0360)	(0.0360)	(0.0360)	(0.0458)	(0.0461)	(0.0461)
Age^2^	0.00120 ***	0.000700 **	0.000699 **	0.00163 ***	0.00100 **	0.00100 **
	(0.000330)	(0.000331)	(0.000331)	(0.000417)	(0.000420)	(0.000420)
Female	0.323 **	0.254 **	0.254 **	0.546 ***	0.487 ***	0.488 ***
	(0.130)	(0.129)	(0.129)	(0.184)	(0.181)	(0.181)
Political connection			0.0485			−0.0330
			(0.0692)			(0.0768)
ROA		0.552	0.544		−0.253	−0.251
		(0.358)	(0.358)		(0.430)	(0.430)
Ln (size)		−0.165 ***	−0.166 ***		−0.179 ***	−0.179 ***
		(0.0129)	(0.0130)		(0.0147)	(0.0147)
Firm age		−0.0285 *	−0.0280 *		−0.0180	−0.0183
		(0.0166)	(0.0166)		(0.0181)	(0.0181)
(Firm age)^2^		0.000946 *	0.000932 *		0.000532	0.000540
		(0.000513)	(0.000513)		(0.000566)	(0.000567)
Education				−0.139 *	−0.0147	−0.0149
				(0.0841)	(0.0858)	(0.0858)
Constant	4.879 ***	6.812 ***	6.808 ***	6.634 ***	8.248 ***	8.246 ***
	(0.978)	(0.998)	(0.999)	(1.249)	(1.269)	(1.268)
Year dummy	Yes	Yes	Yes	Yes	Yes	Yes
Observations	4529	4529	4529	3493	3493	3493

*Note*: * *p* < 0.10, ** *p* < 0.05, *** *p* < 0.01.

**Table 6 ijerph-19-14495-t006:** Chairperson characteristics and environmental investment in production.

	(1)	(2)	(3)	(4)	(5)	(6)
Variables	Ln (inv)	Ln (inv)	Ln (inv)	Inv/Asset	Inv/Asset	Inv/Asset
Age	0.649 ***	0.303 **	0.304 **	−0.0308	−0.0342 *	−0.0344 *
	(0.151)	(0.139)	(0.139)	(0.0194)	(0.0199)	(0.0199)
Age^2^	−0.00558 ***	−0.00282 **	−0.00282 **	0.000308 *	0.000336 *	0.000338 *
	(0.00136)	(0.00125)	(0.00125)	(0.000175)	(0.000179)	(0.000179)
Female	−1.993 ***	−0.880 *	−0.880 *	−0.0416	−0.0323	−0.0324
	(0.557)	(0.513)	(0.513)	(0.0719)	(0.0731)	(0.0731)
Political connection			0.0642			−0.0259
			(0.234)			(0.0333)
ROA		−3.168 ***	−3.186 ***		0.0509	0.0582
		(1.162)	(1.165)		(0.166)	(0.166)
Ln (size)		0.680 ***	0.679 ***		0.00565	0.00601
		(0.0437)	(0.0439)		(0.00624)	(0.00625)
Firm age		0.0805	0.0812		0.00252	0.00223
		(0.0572)	(0.0573)		(0.00816)	(0.00817)
(Firm age)^2^		−0.00189	−0.00192		−0.000178	−0.000167
		(0.00175)	(0.00176)		(0.000250)	(0.000250)
Constant	−10.81 ***	−16.51 ***	−16.52 ***	0.783	0.747	0.751
	(4.161)	(3.926)	(3.928)	(0.537)	(0.560)	(0.560)
Year dummy	Yes	Yes	Yes	Yes	Yes	Yes
Observations	1148	1148	1148	1148	1148	1148
R-squared	0.037	0.213	0.213	0.011	0.015	0.015

*Note*: * *p* < 0.10, ** *p* < 0.05, *** *p* < 0.01.

**Table 7 ijerph-19-14495-t007:** Robustness check.

	(1)	(2)	(3)
Variables	Inv_Type	Ln (inv)	Inv/Asset
Age	−0.152 ***	0.311 **	−0.0348 *
	(0.0474)	(0.139)	(0.0199)
Age^2^	0.00132 ***	−0.00289 **	0.000341 *
	(0.000431)	(0.00125)	(0.000179)
Female	0.455 **	−0.924 *	−0.0300
	(0.187)	(0.514)	(0.0733)
Political connection	−0.0585	0.0754	−0.0266
	(0.0779)	(0.234)	(0.0334)
ROA	0.0738	−3.299 ***	0.0644
	(0.438)	(1.167)	(0.166)
Ln (size)	−0.202 ***	0.690 ***	0.00544
	(0.0152)	(0.0444)	(0.00633)
Firm age	0.0164	0.0567	0.00358
	(0.0186)	(0.0596)	(0.00850)
(Firm age)^2^	−0.000528	−0.00123	−0.000205
	(0.000583)	(0.00182)	(0.000259)
Industry	0.553 ***	−0.262	0.0144
	(0.0565)	(0.175)	(0.0250)
Education	−0.0486		
	(0.0862)		
Constant	9.310 ***	−16.70 ***	0.761
	(1.306)	(3.927)	(0.560)
Year dummy	Yes	Yes	Yes
Observations	3493	1148	1148
R-squared		0.215	0.015

*Note*: * *p* < 0.10, ** *p* < 0.05, *** *p* < 0.01.

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
