# Peer review of "Do the Age and Gender of Chairperson Affect Firms’ Environmental Protection Investment?"

_ijerph, 2022, doi:10.3390/ijerph192114495_

Round 1
Reviewer 1 Report
1. Can add to introduction - that 'in China' can also be considered /contrasted with in comparison to China's footprint impact in other countries (outside).
2. There is some repetition in parts. Can clean.
3. May discuss more the role of Chairperson within board (in China context) and link this to ESG strategy and decision-making more explicitly.
4. The Chairman characteristics are from secondary data (reports) rather than interviews. Is this a limitation of the study? Is this derived from the published reports only or website? Maybe need to qualify as publically available attributes of Chair?
5. Maybe add a bit more about methodology and approach for reader.
5. Can add a bit more about China context.
Author Response
We would like to thank the editor and referee for providing us valuable comments. They help us to further sharpen our argumentation and address some potential misinterpretations. Our detailed, point-by-point responses to the editorial and referees’ comments are given below.
Respond to Reviewer 1
1. Can add to introduction - that 'in China' can also be considered /contrasted with in comparison to China's footprint impact in other countries (outside).
Thanks for your kindly suggestion. We add this information in introduction.
2. There is some repetition in parts. Can clean.
Thanks for your good review. We check our paper and make it clean.
3. May discuss more the role of Chairperson within board (in China context) and link this to ESG strategy and decision-making more explicitly.
Thank for your professional advice. We add some information to discuss the role of Chairperson in China.
4. The Chairman characteristics are from secondary data (reports) rather than interviews. Is this a limitation of the study? Is this derived from the published reports only or website? Maybe need to qualify as publically available attributes of Chair?
Thank for your good comments. In the paper, we focus on age and gender information of chair, this information is publicly available.
5. Maybe add a bit more about methodology and approach for reader.
Thanks for your good advice. We add the methodology section.
6. Can add a bit more about China context.
Thanks for your kindly comments. We add more context about China.
Please refer to the red part of the revised draft for details. Thanks!
Reviewer 2 Report
The article is interesting and generally, it deserves to be published with some revisions that are suggested below:
1. You may try to re-formula the form, for example, section 3(Empirical results), which can be divided into 3. Methodology and Material,4. Results and Discussion,4.1. Statistical Tests, 4.2. Results Analysis, and 4.3. Discussion of the Results, since the paper lack logic.
2. Would you describe how many samples were used, why only have the ones from 2010-2017, and not added more lately samples from 2018-2020 from the CSMAR database?
3. Would you please refer to a more lately journal (2019-2021) since your reference is most of older?
Author Response
We would like to thank the editor and referee for providing us valuable comments. They help us to further sharpen our argumentation and address some potential misinterpretations. Our detailed, point-by-point responses to the editorial and referees’ comments are given below.
Respond to Reviewer 2
1. You may try to re-formula the form, for example, section 3(Empirical results), which can be divided into 3. Methodology and Material,4. Results and Discussion,4.1. Statistical Tests, 4.2. Results Analysis, and 4.3. Discussion of the Results, since the paper lack logic.
Thanks for your professional suggestions. We reorganize our paper as your advice.
2. Would you describe how many samples were used, why only have the ones from 2010-2017, and not added more lately samples from 2018-2020 from the CSMAR database?
Thanks for your good comments. The data samples cover 2010-2017, the main reason we do not have the data after 2017. And we think after 2017, the government encourage more firm to disclose report, and more powerful supervise on firm pollution. Before 2017, the data cover more self-report.
3. Would you please refer to a more lately journal (2019-2021) since your reference is most of older?
Thanks for your good advice. We update the literature review.
Please refer to the red part of the revised draft for details. Thanks!
Reviewer 3 Report
Below are some notes on the paper.
1. The title of the article does not fully reflect the content - it should be stated directly that it is about respondents' sex and age. In addition, the results of the basania do not entitle to such an unequivocal formulation of the title.
2. The literature review contains unnecessary data (eg Figure 2 adds little to the article).
3. The introduction, on the other hand, does not include the formulation of hypotheses. One should derive hypotheses from the literature. Here we need a deeper analysis of the literature and research to date. Attention should be paid to the global studies of women as CEOs.
4. Nothing has been written about the research procedure, nor about the research materials and methods. Suddenly the presentation of the results begins.
5. Paragraph Discussion is missing.
6. There are no theoretical and practical implications of the research.
7. There are no proposals for further research.
8. There are no described research limitations. This is an important part of scientific articles.
9. The literature contains few items.
Author Response
We would like to thank the editor and referee for providing us valuable comments. They help us to further sharpen our argumentation and address some potential misinterpretations. Our detailed, point-by-point responses to the editorial and referees’ comments are given below.
Respond to Reviewer 3
1. The title of the article does not fully reflect the content - it should be stated directly that it is about respondents' sex and age. In addition, the results of the basania do not entitle to such an unequivocal formulation of the title.
Thanks for your good review. We revise the new title “Do the Age and Gender of Chairperson Affect firms’ Environmental Investment?” to directly show our main finding.
2. The literature review contains unnecessary data (eg Figure 2 adds little to the article).
Thanks for your kindly review. In Figure 2, we want to show the spatial distribution of CSR.
3. The introduction, on the other hand, does not include the formulation of hypotheses. One should derive hypotheses from the literature. Here we need a deeper analysis of the literature and research to date. Attention should be paid to the global studies of women as CEOs.
Thanks for your good advice. We add the studies on women CEOs and update the literature in review. And based on literature, we derive one hypothesis.
4. Nothing has been written about the research procedure, nor about the research materials and methods. Suddenly the presentation of the results begins.
Thanks for your good advice. We reorganize the paper, add the research materials and methods section.
5. Paragraph Discussion is missing.
Thank for your good review. We add discussion section.
6. There are no theoretical and practical implications of the research.
Thanks for your good advice. We add new discussion about implications of the paper in discussion section.
7. There are no proposals for further research.
Thanks for your good comment. We add discussion further research.
8. There are no described research limitations. This is an important part of scientific articles.
Thanks for your good review. We add discussion on our research limitation in the last section.
9. The literature contains few items.
Thanks for your detailed suggestion. We reorganize our paper structure, make the paper clearly to read.
Please refer to the red part of the revised draft for details. Thanks!
Round 2
Reviewer 3 Report
The revised version of the article is factually correct enough.